# The Evolving Pathways of the Efficacy of and Resistance to CDK4/6 Inhibitors in Breast Cancer

**DOI:** 10.3390/cancers15194835

**Published:** 2023-10-02

**Authors:** Inês Gomes, Catarina Abreu, Luis Costa, Sandra Casimiro

**Affiliations:** 1Luis Costa Lab, Instituto de Medicina Molecular, Faculdade de Medicina de Lisboa, Universidade de Lisboa, 1649-028 Lisbon, Portugal; ines.gomes@medicina.ulisboa.pt; 2Oncology Division, Hospital de Santa Maria—Centro Hospitalar Universitário Lisboa Norte, 1649-028 Lisbon, Portugal; catarina.abreu@chln.min-saude.pt

**Keywords:** cyclin-dependent kinase 4 and 6 inhibitors (CDK4/6i), breast cancer (BC), therapeutic strategies

## Abstract

**Simple Summary:**

Nowadays, the upfront treatment for patients facing a diagnosis of advanced luminal breast cancer (BC) is a combination of endocrine therapy (ET) with an inhibitor of CDK4/6 (CDK4/6i), which effectively targets and prevents cell cycle progression in hormone-dependent BC. However, the identification of companion predictive biomarkers and ways to overcome or delay the almost inevitable acquired resistance would increase the clinical benefit of this treatment. In this review, we discuss the state-of-the-art evidence about the efficacy of and resistance to CDK4/6i, pinpointing the most relevant past, present and emerging preclinical and clinical efforts.

**Abstract:**

The approval of cyclin-dependent kinase 4 and 6 inhibitors (CDK4/6i) in combination with endocrine therapy (ET) has remarkably improved the survival outcomes of patients with advanced hormone receptor-positive (HR+) breast cancer (BC), becoming the new standard of care treatment in these patients. Despite the efficacy of this therapeutic combination, intrinsic and acquired resistance inevitably occurs and represents a major clinical challenge. Several mechanisms associated with resistance to CDK4/6i have been identified, including both cell cycle-related and cell cycle-nonspecific mechanisms. This review discusses new insights underlying the mechanisms of action of CDK4/6i, which are more far-reaching than initially thought, and the currently available evidence of the mechanisms of resistance to CDK4/6i in BC. Finally, it highlights possible treatment strategies to improve CDK4/6i efficacy, summarizing the most relevant clinical data on novel combination therapies involving CDK4/6i.

## 1. Introduction

The cell cycle is a well-preserved process, tightly regulated and essential for organism development and the maintenance of homeostasis. In the 1970s, the fundamental role of cyclin-dependent kinases (CDKs) in cell cycle biology was unravelled [1,2,3]; a Nobel Prize was later awarded to Leland H. Hartwell, R. Timothy Hunt and Paul M. Nurse for this discovery, in 2001. The CDKs, a family of serine/threonine kinases, form different complexes with cyclins during the cell cycle, regulating cell cycle transition, progression and arrest, as specifically reviewed by [4,5].

In eukaryotic cells, the cell cycle entry is mainly controlled by the proteins CDK4/6, which respond to numerous growth regulatory signals, such as mitogenic, hormonal and growth factor signals, and trigger cell cycle progression from G0/G1 to S phase. CDK4 and CDK6 bind to the D-type cyclins (cyclin D1, D2, or D3), forming active cyclin–CDK complexes that phosphorylate, among other proteins, the retinoblastoma protein (Rb), allowing the release of the E2F transcription factor, which regulates genes involved in the promotion of the cell cycle transition from G1 to S phase. E2F targets include cyclin E1 (CCNE1) and cyclin E2 (CCNE2), which bind to and activate CDK2. This G1/S checkpoint complex leads to Rb hyperphosphorylation and promotes S-phase entry and DNA replication. Cell cycle progression can be suppressed naturally by two families of CDK inhibitors, INK4 (p16INK4A, p15INK4B, p18INK4C and p19INK4D) and CIP/KIP (p21Waf1/Cip1, p27Kip1 and p57Kip2), which together prevent inappropriate cell division. The eukaryotic cell cycle and its major players and regulators are illustrated in Figure 1.

Although the cell cycle is a highly regulated process, it is not a failsafe. In fact, cell cycle dysregulation favoring sustained proliferative signaling is one of the first described hallmarks of cancer [6]. Abnormalities in CDK-related pathways occur in most cancers, and given the critical role of CDKs in cell proliferation, it is unsurprising that CDK inhibition was recognized as an attractive target for anticancer therapy.

The cyclin D1–CDK4/6 axis, in particular, plays a key role in mammary gland biology and breast cancer (BC). Cyclin D1 is required for the proliferation of mammary epithelial cells during pregnancy [7,8], and the knockout of either cyclin D1 or CDK4 prevents the development and growth of mammary carcinomas arising from luminal epithelial cells in mice [9,10,11]. Additionally, dysregulation of the cyclin D–CDK4/6 axis is observed in the majority of hormone receptor-positive (HR+) breast tumors. Estrogen and progesterone receptor (ER and PR, respectively) signaling pathways are mitogenic in HR+ tumor cells by regulating cyclin D1 and increasing its activity, which results in exacerbated cell proliferation through the cyclin D1–CDK4/6 axis [12,13]. The recognition of the prominent role of the cyclin D–CDK4/6 complex in HR+ BC and the fact that the Rb pathway generally remains functional in these tumors made CDK4/6 inhibitors (CDK4/6i) particularly attractive in this subset of BC [14].

Currently, CDK4/6i are the standard-of-care therapy in combination with endocrine therapy (ET) in patients with HR+, human epidermal growth factor receptor 2 (HER2)-negative metastatic BC [15]. Although the approval of CDK4/6i changed the treatment landscape for these patients, 10–20% of the patients turn out to be intrinsically resistant to this therapy, and acquired resistance eventually occurs in virtually all patients [16,17,18,19]. Therefore, the mechanisms underlying resistance to CDK4/6i and the development of new therapeutic strategies to circumvent such resistance is a hot topic in cancer research.

Clinically, primary (intrinsic) endocrine resistance is defined as disease progression during the first six months of first-line endocrine therapy for advanced or metastatic breast cancer, whereas secondary (acquired) endocrine resistance refers to disease progression after six months of treatment [15]. Since CDK4/6i are used in combination with ET, the distinction between resistance to CDK4/6i and endocrine resistance is not straightforward, and the last, per se, is outside the scope of this work. In this review, we will discuss the biological and clinical features of the currently approved CDK4/6i and the mechanisms of resistance identified so far. We want to emphasize that most of the resistance mechanisms known so far are related to both intrinsic and acquired resistance, and it is particularly difficult to clearly implicate the cellular aspects in one or the other. Finally, new therapeutic strategies under clinical investigation to delay or overcome resistance to CDK4/6i are highlighted.

## 2. Selective CDK4/6i

As already mentioned, cell cycle dysregulation towards uncontrolled cancer cell proliferation is often observed across different tumor types. Although the development of CDK inhibitors (CDKi) is a longstanding aspiration in oncology, the development of safe and effective CDKi proved to be difficult for many years [14,20,21], in part because of the biological complexity of the cell cycle but also because cyclin–CDK complexes often exhibit redundancy and plasticity, particularly in cancer cells, which often harbor genetic aberrations in key cell cycle genes.

The first-generation (e.g., flavopiridol, roscovitine) and second-generation (e.g., dinaciclib, SNS-032) CDKi were potent nonspecific pan-CDKi that ultimately failed to demonstrate efficacy in clinical trials, with a low therapeutic index and high-toxicity profiles [20,22]. Notwithstanding these obstacles, third-generation CDKi that selectively inhibit CDK4/6 were developed, leading to a breakthrough in modern BC treatment. Currently, three CDK4/6i are approved for advanced HR+HER2− BC treatment—palbociclib, ribociclib and abemaciclib [23,24,25], now being studied in other BC subtypes, including HR+/HER2+ (e.g., NCT03709082; NCT03913234; NCT02448420), HER2+ (e.g., NCT04351230; NCT02448420) and triple-negative BC (TNBC) (e.g., NCT05067530; NCT03130439).

### 2.1. Selective CDK4/6i Approved to Treat Breast Cancer

The backbone treatment for HR+/HER2− BC continues to be ET, such as aromatase inhibitors (AI), fulvestrant and tamoxifen. However, acquired resistance to ET is near inevitable. Many mechanisms of acquired endocrine resistance were described and were recently reviewed by us and others [26,27,28]. Stemming from several preclinical and clinical studies showing that CDK4/6i act synergistically with ET and can overcome ET resistance [29,30,31,32], a series of randomized trials, namely PALOMA [33,34,35,36,37], MONALEESA [31,38,39,40,41,42] and MONARCH [43,44,45,46,47], showed that CDK4/6i increase progression-free survival (PFS) compared with ET alone in HR+/HER2− BC. Additionally, the MONALEESA and MONARCH clinical trials also showed an improvement in overall survival (OS). These findings led to the approval by the USA Food and Drug Administration (FDA) and European Medicines Agency (EMA) of the three CDK4/6i—palbociclib, ribociclib and abemaciclib—in combination with ET, as first-line and second-line treatments of metastatic HR+/HER2− BC. The administration route, structure and mechanisms of action are similar for the three CDK4/6i, but there are some differences in substrate selectivity and pharmacodynamics, as described below and summarized in Table 1.

#### 2.1.1. Palbociclib

Palbociclib was the first of the CDK4/6i to receive FDA approval, in February 2015, to treat postmenopausal women with metastatic BC [24], based on the results of two phase II clinical trials, PALOMA-1 (NCT00721409) and PALOMA-2 (NCT01740427). Both trials evaluated the efficacy of 125 mg palbociclib in combination with 2.5 mg letrozole (AI) every day (QD) compared to 2.5 mg letrozole QD alone. In PALOMA-1, which included 165 patients, a median PFS of 20.2 months was observed in patients treated with the combination versus 10.2 months with letrozole in monotherapy (HR 0.488, 95% CI 0.319–0.748, one-sided *p* = 0.0004) [18]. The prolonged PFS in patients treated with palbociclib and ET was further confirmed in the PALOMA-2 trial, where 666 patients were included, with a median PFS of 24.8 versus 14.5 months when comparing the combination with letrozole alone (HR 0.58, 95% CI 0.46–0.72, *p* < 0.001) [30].

The PALOMA-3 phase III clinical trial (NCT01942135) led to the subsequent approval of palbociclib in combination with fulvestrant, regardless of menopausal status. This trial evaluated the efficacy of 125 mg QD palbociclib (3 weeks on/1 week off) with 500 mg fulvestrant (q4w) versus 500 mg fulvestrant q4w in 521 metastatic HR+/HER2− BC patients. In this study, an increase in median OS from 28.0 months to 34.9 months was observed in the combination group (HR 0.81, 95% CI 0.64–1.03, *p* = 0.09) [36]. Although a significant increase in OS was not reported in PALOMA trials, there was a clinical benefit of palbociclib in HR+/HER2− BC patients.

The mean half-life of palbociclib is 29 ± 5 h, and due to myelosuppressive effects, it is dosed daily for 21 days followed by a one-week break to enable neutrophil count recovery [48]. The adverse effects more frequently observed in patients are neutropenia, leukopenia and fatigue, which are generally manageable and reversible [18,30,36].

Palbociclib has also been studied in other malignancies, as recently reviewed [49], but its clinical use remains a niche for HR+/HER2− BC.

#### 2.1.2. Ribociclib

Ribociclib was the second of the CDK4/6i to obtain FDA approval, in March 2017 [23]. The specificity and pharmacodynamics profiles of ribociclib are very similar to palbociclib (Table 1). The mean half-life is 32 h and the administration schedule is the same as palbociclib, dosed daily for 21 days followed by a one-week break [48]. The adverse events most observed in patients include neutropenia, hepatotoxicity and corrected QT interval (QTc) prolongation, which can also be controlled by dose interruption, dose reduction and additional symptomatic supportive treatment [31,38,39].

The phase III clinical trials MONALEESA-2 (NCT01958021) and MONALEESA-7 (NCT02278120) led to the approval of ribociclib plus AI as a first-line treatment for metastatic HR+/HER2− BC patients regardless of menopausal status. In MONALEESA-2, the efficacy of 600 mg ribociclib plus 2.5 mg letrozole QD was compared with 2.5 mg letrozole QD alone in 668 postmenopausal women, with PFS improving to a median of 25.3 months in the combination arm versus 16.0 months in the single arm (HR 0.57, 95% CI 0.46–0.70, *p* < 0.001) [50]. Median OS was also significantly prolonged (63.9 vs. 51.4 months, (HR 0.76, 95% CI 0.63–0.93, two-sided *p* = 0.008) [40]). The MONALEESA-7 included 672 premenopausal or perimenopausal women and replicated the results of MONALEESA-2, with a median PFS of 23.8 months in the combination arm compared with 13.0 months in the single arm (HR 0.55, 95% CI 0.44–0.69, *p* < 0·0001) [39], and an improvement in OS at 42 months from 46% to 70.2% (HR 0.71, 95% CI 0.54–0.95, *p* = 0.00973) [41].

Subsequently, the MONALEESA-3 phase III clinical trial (NCT02422615) allowed the approval of the combination of 600 mg ribociclib QD plus 500 mg fulvestrant q4w in 726 postmenopausal women as an initial endocrine-based therapy or after progression on ET, reporting also an improved median PFS when patients were treated with ribociclib plus ET versus ET alone (20.5 vs. 12.8 months; HR 0.593, 95% CI 0.480–0.732, *p* < 0.001) [38], and an improvement in OS at 42 months from 45.9% to 57.8% (HR 0.72, 95% CI 0.57–0.92, *p* = 0.00455) [42].

#### 2.1.3. Abemaciclib

The last of the CDK4/6i approved by the FDA was abemaciclib, in February 2018 [51]. Although all three CDK4/6i are considered selective CDK4/6i, several studies, reviewed by other authors [48,52], demonstrated that abemaciclib also inhibits CDK9 (Table 1). The mean half-life of abemaciclib is also considerably different (18.3 h), as is its treatment administration scheme. Since abemaciclib induces less bone marrow suppression, it can be given twice per day continuously without a break [48]. The most frequent side effects are also different; diarrhea and fatigue are the most commonly observed [43,47,53].

Abemaciclib was first approved in combination with fulvestrant based on the MONARCH-2 (NCT02107703) phase III clinical trial, which enrolled 669 patients [47]. Patients with HR+/HER2− advanced BC who progressed under ET were given either 150 mg abemaciclib twice daily (Q12H) plus 500 mg fulvestrant on days 1 and 15 of the first cycle and subsequently q4w or fulvestrant alone in the same regime as in the combination group, showing an improvement in median PFS of 16.4 months versus 9.3 months (HR 0.553, 95% CI 0.449 to 0.681, *p* < 0.001). There was also an increase in OS from 37.3 months to 46.7 months (HR 0.757, 95% CI 0.606–0.945, *p* = 0.0137) [46].

Subsequently, abemaciclib was also approved in combination with AIs based on the MONARCH-3 (NCT02246621) phase III clinical trial [43]. The study enrolled 493 postmenopausal HR+/HER2− BC patients previously treated with ET, and 1 mg anastrozole or 2.5 mg letrozole alone QD or combined with 150 mg abemaciclib Q12H were compared. The median PFS in the single-therapy arm was 14.7 months, while the median PFS of the abemaciclib group reached 28.1 months (HR 0.540, 95% CI 0.418–0.698; *p* = 0.000002) [17]. A recent interim analysis presented at ESMO Congress 2022 showed the benefit in OS at 70.2 months median follow-up, with a median OS of 67.1 months for abemaciclib + NSAI vs. 54.5 months for placebo + NSAI (HR 0.754, 95% CI 0.584–0.974, 2-sided *p* = 0.0301) [54].

So far, abemaciclib is the only one of the CDK4/6i approved for HR+/HER2− advanced or metastatic BC as a monotherapy, based on results from the MONARCH-1 (NCT02102490) phase II clinical trial. A total of 132 poor prognosis and heavily pretreated patients with refractory HR+/HER2− BC who had progressed on prior ET and had received at least two prior chemotherapy regimens were enrolled, and 200 mg abemaciclib or a placebo was given Q12H. The average PFS was 6.0 months (95% CI 4.2–7.5), and the median OS was 17.7 months (95% CI 16.0–NR) [53].

Due to its demonstrated efficacy in advanced BC, the use of CDK4/6i is also being studied in early-stage BC, in both neoadjuvant and adjuvant settings (Table 2). The first and so far only approval was in 2022, for abemaciclib in combination with ET for adjuvant treatment of early HR+/HER2− BC at high risk of recurrence [51].

Based on the clinical success of therapeutically approved CDK4/6i in HR+/HER2− metastatic BC and on encouraging preclinical results, several new selective CDK4/6i are currently being investigated to treat BC (Table 1), as recently reviewed [49,63].

For example, trilaciclib (G1T28) was approved in 2021 to reduce chemotherapy-induced bone marrow suppression in patients with advanced-stage small cell lung cancer [64]; furthermore, a recent phase III clinical trial (PRESERVE 2, NCT04799249) is evaluating its efficacy in TNBC [65]. Dalpiciclib (SHR-6390) in combination with ET is also in phase III clinical trial (DAWNA-1, NCT03927456) for patients with HR+/HER2− BC [66] and in phase I/II trials for several BC subtypes in combination with ET, chemotherapy or immunotherapy [67].

Overall, the search for new selective CDK4/6i is an expanding field in oncology, and the use of CDK4/6i beyond HR+/HER2− BC is under lively investigation. In the next sections, we will review the mechanism of action behind the therapeutic efficacy of CDK4/6i as well as the main discoveries and evidence supporting the so-far postulated mechanisms of resistance.

### 2.2. Mechanism of Action of CDK4/6i

Despite cancer cell cycle arrest being the most obvious cellular response to CDK4/6i, the effects of CDK4/6i seem to be more complex than initially thought [68]. The widespread use of these compounds in preclinical research and clinical trials has provided compelling evidence that CDK4/6i affect several cellular characteristics across different cell types like tumor, immune and stromal cells (Figure 2).

#### 2.2.1. Effect on Tumor Cells

CDK4/6i target the ATP-binding pocket of CDK4 and CDK6, preventing downstream CDK4/6-mediated phosphorylation of Rb [69,70]. Consequently, the best-characterized effect of CDK4/6i is the proliferative arrest in G1 due to the inhibition of E2F transcriptional activity [11,29,32,70]. Such a cytostatic effect is expected to lead to tumor growth stabilization; however, tumor shrinkage was also reported in CDK4/6i monotherapy trials [53,71], suggesting additional cellular effects. In fact, alongside proliferation, E2F targets also regulate processes such as DNA repair [72], DNA methylation [73], chromatin condensation [74], cellular metabolism [75] and apoptosis [76,77], supporting the hypothesis that CDK4/6i may affect any of these processes in Rb-proficient cells [74,78,79,80,81,82].

Multiple preclinical studies reported that CDK4/6i can induce a senescence-like state in cancer cells characterized by cellular enlargement and increased senescence-associated β-galactosidase (SAβGal) activity [83,84,85,86,87,88,89]. Two studies showed that this senescent-like state seems to be mostly Rb-dependent [90,91], which is not surprising since Rb is one of the key mediators of cellular senescence, as recently reviewed [92], but might also be linked to reduced activity of the direct CDK4/6 substrates forkhead box protein M1 (FOXM1) [93,94] and DNA methyltransferase 1 (DNMT1) [94].

It was also described that CDK4/6i affect chromatin remodeling in an Rb-dependent manner, for example, by inducing alpha-thalassemia mental retardation X-linked protein (ATRX) expression and proteolytic degradation of mouse double minute 2 homolog (MDM2), which ultimately also promotes senescence [95]. Additionally, CDK4/6i can lead to widespread enhancer activation [74], directly implicated in apoptotic evasion and enhanced cellular immunogenicity, mainly via activator protein 1 (AP-1) transcription factor. AP-1 is known to be involved in classical senescence, where it drives chromatin accessibility and enhancer activation [89,96,97].

Autophagy and senescence are closely connected and often regulated by similar signaling pathways. It was already reported that CDK4/6i increased autophagy markers in preclinical HR+ BC models and that the use of autophagy inhibitors further enhanced the senescent phenotype of CDK4/6i-treated BC cells [98]. Further studies are necessary to elucidate the role of CDK4/6i in autophagy.

Since cell division is coordinated with the cellular metabolic state, a possible effect of CDK4/6i in tumor metabolism also started to emerge. Some studies reported that CDK4/6i can induce tumor cell metabolic reprogramming and have an impact on mitochondria [89,99], lysosomes [93,100,101] and glycolysis [102,103]. How this metabolic modulation affects therapy response is still to be clarified.

Lastly, CDK4/6 inhibition was shown to upregulate programmed death-ligand 1 (PD-L1) in tumor cells [104,105].

#### 2.2.2. Effect on Immune Cells

In the last years, the impact of CDK4/6i on the immune system gained attention. Several preclinical studies described that CDK4/6i themselves could boost antitumor immune responses in BC and other cancers, driven by T cell-intrinsic mechanisms as well as enhancement of antigen presentation in tumor cells. Regarding T cells, it was shown that treatment with CDK4/6i could repress regulatory T cells (Treg) proliferation [79,106,107,108], activate CD8+ T cells [79,106,109,110,111] and increase infiltration of effector T cells (Teff) on tumor microenvironment (TME) via the upregulation of the nuclear factor of activated T cells (NFAT) signaling [104,107].

Additionally, treatment with CDK4/6i enhances the secretion of immune-stimulatory chemokines like inflammatory chemokines C-C motif ligand 5 (CCL5), C-X-C motif ligand (CXCL), CXCL9 and CXCL10 [89,107,111]. CDK4/6i also prompt hypomethylation and expression of endogenous retroviruses (ERVs), therefore inducing a double-stranded RNA (dsRNA) response [79]. This, in turn, leads to increased expression and secretion of type III interferon (IFN), activation of Janus kinase (JAK)–signal transducer and activator of transcription (STAT) signaling, increased IFN-driven gene expression and upregulation of major histocompatibility complex (MHC) class I [79,104]. Lastly, CDK4/6i-driven chromatin remodeling facilitates IFN-mediated expression of IFN-responsive genes [74].

Given these observations, some clinical trials were designed to leverage the effect that CDK4/6i have in producing antitumor immune responses to develop new clinically effective therapies. For example, the efficacy of CDK4/6i in combination with anti-PD-1 or anti-PD-L1 antibodies is currently being tested and will be revised below.

#### 2.2.3. Effect on Other Cell Types

Most studies with CDK4/6i mainly focused on their effect on cancer cells and, more recently, on immune cells. Nevertheless, it is not surprising that CDK4/6i may also affect other cellular components within the TME or more broadly.

CDK4/6 regulate both proliferation and senescence of fibroblasts [112,113]. More recently, it was demonstrated that treatment of fibroblasts with either palbociclib or abemaciclib induced a senescent phenotype, characterized by the secretion of a large number of proinflammatory cytokines, which may have an impact on tumor cells by suppressing antitumor responses [83,114].

Additionally, endothelial cell proliferation and angiogenesis are CDK4/6-dependent [115,116]; it was shown that treatment of endothelial cells with palbociclib resulted in cell cycle arrest [117]. However, the consequences of this effect on angiogenesis and tumor outcomes are still unknown.

More studies on the effect of CDK4/6 inhibition on other cell types are clearly needed to understand better the effect of CDK4/6i in the interaction between host and cancer cells.

## 3. Mechanisms of Resistance to CDK4/6i and Possible Strategies to Overcome Them

Until now, multiple mechanisms of resistance to CDK4/6i have been described and several predictive biomarkers have been proposed, despite none of them being successfully validated for clinical use. Increasing the knowledge about the molecular mechanisms involved in CDK4/6i resistance is extremely important and will certainly contribute to the development of new therapeutic strategies to circumvent resistance or increase benefit as well as potentially disclose the so needed predictive biomarker(s). In this section, we will review the several mechanisms of resistance described so far, which can be implicated in both intrinsic and acquired resistance, as well as the novel therapeutic combinations currently in clinical trials that intend to overcome or delay such resistance (summarized in Table 3).

### 3.1. Mechanisms of Resistance to CDK4/6i

#### 3.1.1. Retinoblastoma Protein (Rb)

The tumor suppressor Rb protein is the main target of the cyclin D–CDK4/6 complex, and, as already mentioned, Rb-proficiency is expected to affect the efficacy of CDK4/6i. Thus, loss or inactivation of Rb protein is inevitably linked to intrinsic CDK4/6i resistance, and even acquired resistance, which was demonstrated in several studies.

In a pivotal study, Finn and colleagues observed that BC cell lines with increased expression of Rb and cyclin D1 and downregulation of p16 at baseline were more sensitive to palbociclib, while Rb loss yielded therapeutic failure [29]. Other studies proceeded to demonstrate that Rb loss of function was associated with CDK4/6i resistance in BC, either in vitro [118,119,120,121], in vivo [122], or using patient-derived xenograft (PDX) BC models [118,123]. Moreover, restoration of RB1 expression was able to restore tumor cells’ sensitivity to CDK4/6i [118]. In accordance with the potential of Rb as a biomarker to predict primary response to CDK4/6i, a signature of Rb loss of function (Rbsig) developed using The Cancer Genome Atlas (TCGA) dataset was able to discriminate the palbociclib-resistant BC cell lines from the sensitive ones [120,121].

Following these preclinical studies, clinical results started to emerge. Genomic analysis of BC patients treated with CDK4/6i revealed that loss of *RB1* was associated with treatment resistance and worse prognosis [130,137]. However, tumor analyses performed on PALOMA-2 and -3 clinical trials failed to show a significant correlation between Rb expression and intrinsic resistance to CDK4/6i [124,125].

Additionally, analysis of circulating tumor DNA (ctDNA) from BC patients that had disease progression on CDK4/6i revealed the acquisition of *RB1* loss-of-function mutations in some of these patients [170,171]. However, ctDNA analysis of the PALOMA-3 clinical trial showed that acquisition of such mutations only occurred in 6 of 127 (4.7%) patients [151], suggesting that their contribution to acquired resistance is discrete.

Hence, the predictive value of Rb remains unclear, and further investigation in large clinical sets is needed to determine the frequency of Rb mutations and Rb’s potential use as a predictive biomarker for CDK4/6i.

#### 3.1.2. Cyclin D–CDK4–CDK6 Axis

Given that the primary target of CDK4/6i is the cyclin D–CDK4/6 axis, alterations in the expression of these proteins, driven by gene amplification, mutations and epigenetic changes, are expected to affect therapeutic efficacy.

CDK4 amplification in breast tumors was linked to increased tumor cell proliferation [172], development of distant metastasis and poor clinical outcome [173]. Gene expression analysis of palbociclib-resistant BC cell lines demonstrated an increased expression of *CDK4*, along with other cell cycle-related genes such as *CDK2*, *CDK7* and *CCNE1* [128]. However, in abemaciclib-resistant BC cell lines, *CDK6* and not *CDK4* overexpression was associated with resistance to CDK4/6i, with CDK6 knockdown restoring sensitivity [129].

Other studies highlighted the central involvement of CDK6 overexpression in the development of resistance to CDK4/6i in BC [118,130,131,132,133]. In some of these models, resistance-associated CDK6 overexpression was dependent on or a consequence of alterations in other pathways. For example, one study demonstrated that the increased expression of CDK6 was dependent on the suppression of the transforming growth factor-beta (TGF-β) pathway due to miR-432-5p [132]. Another study showed that FAT atypical cadherin 1 (FAT1) loss of function was responsible for CDK6 overexpression via suppression of the Hippo pathway [130].

The upregulation of the cyclin D family of proteins is also a potential mechanism of CDK4/6i resistance. Amplification of *CCND1* occurs in breast tumors and is associated with an increase in tumor proliferation [174]. The upregulation of cyclin D1 and cyclin D2, along with cyclin A, cyclin E and CDK2, was also observed in CDK4/6i-resistant breast models [118,126].

A recent study reported that cyclin D1 and CDK4 proteins were significantly upregulated in BC cells with acquired resistance to palbociclib, downstream of a hyperactivation of PI3K/mTOR pathway [127]. The use of PI3K/mTOR inhibitors decreased cyclin D1 and CDK4 protein levels and restored the sensitivity to palbociclib in resistant cells, both in vitro and in vivo. However, *CCND1* amplification failed to associate with PFS in PALOMA-1 [18] and PALOMA-3 [124] clinical trials. Overall, despite some evidence of the involvement of the cyclin D–CDK4/6 axis in primary and secondary resistance to CDK4/6i, clinical data that support this hypothesis are still missing.

#### 3.1.3. Cyclin E–CDK2 Axis

The hyperactivation of cyclin E and/or CDK2 may compensate for CDK4/6 loss and bypass CDK4/6 inhibition by hyperphosphorylating Rb, providing a possible mechanism of resistance to CDK4/6i.

In this context, and as mentioned, upregulation of cyclin E1, cyclin E2 and CDK2 were already observed in BC CDK4/6i-resistant models [118,120,129,134,135]. Moreover, silencing of either *CCNE1* or *CDK2* restored the sensitivity of resistant cells to palbociclib [118], suggesting an important role in resistance.

Analysis of clinical data further support that the increase in cyclin E1 can be implicated in resistance to CDK4/6 inhibition. An exploratory analysis from the NeoPalAna clinical trial showed a positive association between high expression of *CCNE1* and resistance to palbociclib [59]. Likewise, analysis of tumors from patients enrolled in the PALOMA-3 clinical trial also showed that high expression of *CCNE1* was associated with early progression and decreased therapy efficacy [124]. However, this correlation was not observed in patients from the PALOMA-2 or MONALEESA-2 clinical trials [50,125].

Despite the promising results from different studies, there are conflicting results. Therefore, additional preclinical and clinical studies will be necessary to clarify and confirm the importance of cyclin E–CDK2 axis as a mechanism of intrinsic and acquired resistance to CDK4/6 inhibition.

#### 3.1.4. CDK7

CDK7 is a CDK-activating kinase (CAK) that forms a complex with cyclin H and metastasis-associated protein MTA1 (MAT1), inducing the phosphorylation of CDK1, CDK2, CDK4 and CDK6, promoting cell cycle progression [175]. CDK7 is also an element of the transcription factor II Human (TFIIH) involved in transcription initiation and DNA repair [176]. Given its role, CDK7 is thought of as a mechanism to bypass G1/S inhibition in CDK4/6i-resistant cells.

Indeed, upregulation of CDK7 was observed in CDK4/6i-resistant cell lines, where the sensitivity to CDK7 inhibitors appears to be associated with the loss of ER and Rb [128]. We also found that CDK7 is upregulated in cell lines with acquired resistance to palbociclib [136]. These observations suggest that CDK7 may play an important role in resistance to CDK4/6i, representing a potential therapeutic target to bypass resistance to this class of inhibitors. Recently, it was reported that the CDK7 selective inhibitor THZ1 could significantly inhibit the proliferation of TNBC tumor cells [177].

#### 3.1.5. INK4 and CIP/KIP Families of CDK Suppressors

The INK4 family is a set of intrinsic tumor suppressor factors that competitively bind to CDK4/6, preventing the formation of the cyclin D–CDK4/6 complex, hence inhibiting cell cycle progression. Although it has been hypothesized that the amplification of these proteins could contribute to CDK4/6i resistance, only p16 (encoded by *CDKN2A*) is implicated in intrinsic resistance to CDK4/6i. It was found that p16 overexpression in Rb-proficient models can decrease CDK4 levels and induce resistance to CDK4/6i [123]. Moreover, a recent study showed that p16 overexpression was associated with reduced antitumor activity of CDK4/6i in BC cell lines, PDXs and patients with advanced BC [137]. However, in the PALOMA-1, -2 and -3 clinical trials, no significant difference was observed in patients harboring *CDKN2A* amplification in terms of PFS and treatment efficacy [124,125].

The CIP/KIP family of CDKi includes p21CIP1 (encoded by *CDKN1A*), p27KIP1 (encoded by *CDKN1B*) and p57KIP2 (encoded by *CDKN1C*), with p21 and p27 being well characterized for their role as negative regulators of G1-phase cell cycle progression. So far, evidence of the potential involvement of CIP/KIP members in CDK4/6i resistance is still very scarce and mostly restricted to p27. It was reported that increased phosphorylation of p27 can inhibit CDK4 and consequently decrease the sensitivity of BC cells to palbociclib [120,138].

#### 3.1.6. Other Cell Cycle Regulators

##### WEE1

WEE1 is a tyrosine kinase protein that phosphorylates and inhibits CDK1 and CDK2. The inhibition of CDK1 helps to maintain the cell in an inactive state and prevent mitosis, whereas inhibition of CDK2 delays the replication process, allowing for DNA repair [178]. Overexpression of WEE1 was shown to induce intrinsic resistance to CDK4/6i, whereas knockdown or pharmacological inhibition of WEE1 were able to restore sensitivity [139,140]. These findings suggest that WEE1 could be a good target for new therapeutic strategies in breast tumors resistant to CDK4/6 inhibition.

##### MDM2-TP53

MDM2 is a protein that negatively regulates the tumor suppressor p53 (TP53); therefore, high activity of MDM2 can prevent DNA repair, leading to improper cell cycle progression of damaged cells [179,180]. MDM2 overexpression is found in about 20% of BC patients [12], which seems to be particularly important in promoting the progression of HR+ BC [179]. The use of an MDM2 inhibitor in combination with palbociclib plus fulvestrant could abrogate resistance to CDK4/6i and ET in a panoply of preclinical models, suggesting that MDM2 may have a role in CDK4/6i resistance, and its inhibition can be a promising therapeutic option in this setting [141].

Although significant enrichment in TP53 mutations was found in tumor samples resistant to CDK4/6i, additional preclinical data showed that baseline sensitivity to CDK4/6i in vitro was similar in TP53 wild-type and TP53-mutant HR+ cell lines [134]. Moreover, TP53 knockout in wild-type cell lines affects sensitivity to palbociclib or abemaciclib in vitro, suggesting that TP53 itself is not sufficient to drive resistance.

##### APC/C-FZR1

Fizzy and cell division cycle 20 related 1 (FZR1) protein is a coactivator of the anaphase-promoting complex (APC), an important regulator of the cell cycle [181]. Through a post-translational mechanism, APC/C-FZR1 interacts with Rb and is a noncanonical regulator of the G1–S transition; its loss can lead to uncontrolled cell cycle progression [182].

It was shown that the knockdown of Rb or FZR1 resulted in a decrease in palbociclib-induced cell cycle arrest in HR+ BC cells, which was stronger in cells with double knockdown, suggesting a synergistic bypass of the cell cycle arrest induced by palbociclib [142]. It would be interesting to further investigate the precise mechanism of intrinsic resistance to CDK4/6i associated with the loss of FZR1 and to validate such a mechanism in patients.

##### AP-1

The AP-1 family consists of homo- and heterodimers of c-Fos, c-Jun, activating transcription factor 1 (ATF) and MAF BZIP transcription factor (MAF), which are transcriptional regulators of several genes, including *CCDN1* [183,184].

It was found that AP-1 and c-Fos increase in BC cells upon acquired resistance to palbociclib, and that AP-1 blockade (by RNA interference) combined with palbociclib could effectively inhibit cell proliferation and simultaneously decrease p-Rb and CDK2 protein levels [143]. Given these results, it would be interesting to explore the effect of c-Fos and/or AP-1 inhibitors in HR+ BC preclinical models of acquired resistance to CDK4/6i.

##### TK1

Thymidine kinase 1 (TK1) is a cytosolic enzyme involved in DNA synthesis and regulation of cell proliferation [185]. TK1 is an E2F target gene proposed as a possible biomarker of CDK4/6i response. It was found that TK1 levels and TK activity (TKa) are reduced after treatment with palbociclib only in sensitive ER+ BC cell lines and that low TKa at baseline measured in plasma samples from patients enrolled in the TREnd trial (NCT02549430) was correlated with longer PFS [145]. Additionally, patients with increasing TKa levels during palbociclib treatment had a shorter time to disease progression, whereas patients with decreased or stable TKa had a better response to palbociclib. In accordance, an association between TK1 overexpression and resistance to palbociclib plus ET was observed in an independent cohort of HR+/HER2− BC patients [144].

Given the promising results, the prognostic value of plasma TKa in patients treated with CDK4/6i plus ET is being assessed in the context of ongoing clinical trials like BioItaLEE (NCT03439046) and PYTHIA (NCT02536742).

##### Aurora Kinase A

Aurora kinase A (encoded by *AURKA*) has a key role in controlling chromosome assembly and segregation during mitosis [186]. The amplification of *AURKA* was found in tumor biopsies from patients resistant to CDK4/6i treatment, while no alterations were detected in samples from CDK4/6i-sensitive patients [134]. Moreover, the accumulation of mitotic errors was reported in a subset of preclinical models of HR+ BC with acquired resistance to palbociclib, in which Aurora kinase A inhibitors had antiproliferative and cytotoxic effects [187].

##### c-Myc

c-Myc is critical to the regulation of many growth-promoting signal transduction pathways and a major driver of cell proliferation [188]. The analysis of ctDNA from patients enrolled in the MONARCH-3 trial showed that *MYC* was one of the top three mutated genes in the abemaciclib plus AI group (*p* = 0.015) [146]. At the preclinical level, it was found that the cyclin E–CDK2-mediated phosphorylation of c-Myc is responsible for resistance to CDK4/6i because it suppresses c-Myc-induced senescence [189]. In accordance with these findings, preclinical data combined with the analysis of clinical transcriptome specimens showed that c-Myc induction and cyclin E/CDK2 activity followed CDK4/6i therapy [147]. Interestingly, in this study, the CDK2/4/6 inhibitor PF-06873600 showed robust preclinical antitumor activity, particularly in the c-Myc activated models.

##### c-MET/FAK

The mesenchymal–epithelial transition factor (c-MET) family receptor was implicated in intrinsic resistance to CDK4/6i via CDK4/6-independent activation of CDK2 by c-MET and its downstream effector focal adhesion kinase (FAK) [190]. Like c-Myc, c-MET was also found to be frequently activated in patients treated with abemaciclib plus AI [146]. This could be important, since the activation of the c-MET/TrkA-B pathways was described as being involved in therapeutic resistance of glioblastoma to CDK4/6i, and c-MET inhibition with altiratinib in combination with CDK4/6i can be effective in this setting [191].

#### 3.1.7. Hormone Receptors and HER2

Given that ER and progesterone receptor (PR) signaling are major promoters of cyclin D–CDK4/6 activity in HR+ BC cells [192], it was not unexpected that loss of ER/PR expression was associated with primary resistance to CDK4/6i in BC cell lines and tumor biopsy specimens [128,129]. However, although the acquisition of mutations in *ESR1* was observed in 25–40% of the patients with advanced HR+/HER2− BC that progressed under ET, particularly with AIs [193,194], ctDNA analysis from patients enrolled in PALOMA-3 revealed a limited predictive value of *ESR1* mutations [151,152].

The expression of androgen receptor (AR) in BC is frequent and occurs in more than 70% of breast tumors [195]. AR activation can be associated with acquired resistance to CDK4/6i, since loss of ER signaling accompanied by activation of AR signaling was observed in palbociclib-resistant BC cell lines [148]. The AR pathway can activate cyclin D1, cyclin E and CDK2, promoting cell cycle progression [128,196,197], thereby decreasing CDK4/6i efficacy. Importantly, the combination of palbociclib with enzalutamide, a selective AR inhibitor, abrogates palbociclib resistance in BC cell lines [148], suggesting that AR can be a new possible target in BC patients resistant to CDK4/6i.

Paradigmatically, HER2-activating mutations were linked to acquired resistance to CDK4/6i in HR+/HER2− BC patients. The analysis of metastatic biopsies from eight BC patients who developed resistance to AI showed the acquisition of HER2 mutations, which conferred estrogen independence as well as resistance to ET and CDK4/6i [149]. In the study, resistance was overcome by combining ET with the irreversible HER2 kinase inhibitor neratinib. Additionally, whole-exome sequencing of 59 tumors treated with CDK4/6i revealed *ERBB2* amplification amongst multiple candidate resistance mechanisms [134]. In another study, the genomic analysis of 24 tumor and 17 ctDNA samples obtained from patients treated with exemestane plus everolimus and palbociclib, triplet therapy for CDK4/6i-resistant BC, found *ERBB2* amplification in several tumors and identified a patient with coexisting tumor lineages with distinct activating *ERBB2* mutations, suggesting that HER2 activation may follow CDK4/6i therapy [150].

#### 3.1.8. PI3K-AKT-mTOR Pathway

The PI3K/AKT/mTOR signaling pathway is frequently altered in BC, mostly from the HR+ subtype [12,153,198], being associated with tumor development, disease progression and therapy resistance, as recently reviewed [199].

Several studies supported the activation of the PI3K/AKT/mTOR pathway as a mechanism to escape CDK4/6i [118,126,153,154,155,156,157,158]. The PI3K/AKT/mTOR pathway was found to be upregulated after prolonged exposure to CDK4/6i or in CDK4/6i-resistant BC models [134,159]. The combination of PI3K-AKT-mTOR and CDK4/6 inhibitors was proven to reduce cell viability and overcome intrinsic and adaptive resistance or even to prevent or delay drug resistance [118,126,155,156,159,160].

Some molecular mechanisms that might lead to PI3K/AKT/mTOR-mediated resistance to CDK4/6i have emerged. For instance, increased levels of phosphorylated AKT induce the formation of the cyclin E2–CDK2 complex, allowing cells to bypass CDK4/6 inhibition [118]. Along with this observation, it was found that activation of 3-phosphoinositide-dependent protein kinase 1 (PDK1) can directly phosphorylate AKT [126,200], upregulating S-phase cyclins and CDKs and mediating acquired resistance to CDK4/6i [126]. In turn, CDK2 can also directly phosphorylate and activate the AKT pathway [157], and the pharmacologic inhibition of PDK1 or CDK2 in combination with a CDK4/6i synergistically induces cell cycle arrest [126,157].

Moreover, not only was it found that mTORC1 activation resulted in increased cyclin D1 expression but also that mTORC1/2 inhibitor could decrease E2F-mediated transcription, suggesting that mTOR upregulation may induce resistance to CDK4/6i via an increase in E2F-mediated transcription [157].

In addition, another study showed that phosphatase and tensin homolog (PTEN) loss, which acts as a negative regulator of the AKT/mTOR signaling pathway, increases the expression of AKT, CDK4 and CDK2 in CDK4/6i-resistant BC cell models [155].

Analysis of *PIK3CA* mutations in plasma ctDNA from patients enrolled in the PALOMA-3 clinical trial revealed an association with worse PFS [151,161], suggesting that *PIK3CA* mutations may be used as a prognostic factor to monitor the efficacy of palbociclib and fulvestrant. Finally, increased levels of activated AKT were recently associated with low PFS in an independent cohort of HR+/HER2− BC patients treated with CDK4/6i and ET [154].

Taken together, the PI3K/AKT/mTOR pathway seems to be associated with CDK4/6i resistance, and blocking this pathway in combination with CDK4/6-targeted therapy represents a promising strategy. Furthermore, the detection of mutations in members of the PI3K/AKT/mTOR pathway as prognostic biomarkers seems promising, and further validation in large cohorts is important.

#### 3.1.9. FGFR Pathway

The fibroblast growth factor receptor (FGFR) signaling pathway comprises a family of receptor tyrosine kinases (FGFR1, FGFR2, FGFR3, FGFR4 and FGFR5) that are involved in proliferation, differentiation and cell survival [201]. The activation of the FGFR pathway has an important role in the development of the normal mammary gland as well as in the development and progression of BC [202].

FGFR1 upregulation was suggested as a potential mechanism of acquired resistance to CDK4/6i [50,128,152,162]. The expression of FGFR1 increased in BC cells with acquired resistance to ET and CDK4/6i, and combination of CDK4/6i with lucitanib or erdafitinib, a FGFR inhibitor, was sufficient to abrogate such resistance [162].

Importantly, *FGFR1* amplification was also found to be associated with CDK4/6i resistance and shorter PFS in patients enrolled in the MONALEESA-2 and PALOMA-3 clinical trials [50,152,162], suggesting that *FGFR1* can also be used as a prognostic marker. Furthermore, FGFR2 mutations were also found in tumors from a different cohort of patients that presented resistance to CDK4/6i [134].

Given these observations, inhibition of the FGFR pathway may be a viable option to overcome resistance to CDK4/6i, and validation in clinical studies is ongoing and will be discussed in the next section.

#### 3.1.10. MAPK-ERK Pathway

The activation of the mitogen-activated protein kinase (MAPK) signaling pathway and several of its downstream effectors, such as RAS-RAF-MEK-ERK, was observed in CDK4/6i-resistant models [128,163]. Indeed, in BC cell lines with acquired resistance to CDK4/6i, the combination of MEK inhibitors with CDK4/6i plus ET was shown to be effective in blocking cell proliferation [162,163], supporting a possible role of the MAPK pathway in CDK4/6i acquired resistance.

Although the oncogenic role of RAS has been extensively demonstrated, the frequency of RAS mutations in human BC has proven to be much lower than expected [203]. However, activating mutations or amplification in *KRAS*, *HRAS* and *NRAS* were found in CDK4/6i-resistant tumors, while no alterations were present in therapy-sensitive counterparts [134]. Recently, the analysis of blood samples from 106 patients with HR+/HER2− metastatic BC treated with palbociclib in combination with fulvestrant as the first-line metastatic therapy showed that the appearance of *KRAS* mutations was associated with palbociclib resistance acquisition within 6 months [204]. This suggests that monitoring the *KRAS* status by liquid biopsy could be used to predict response to CDK4/6i.

#### 3.1.11. RANK-RANKL Pathway

The signaling pathway of the receptor activator of nuclear factor-kB (RANK) and its ligand (RANKL) has a crucial role in several physiological processes, namely, bone remodeling, mammary gland development and functional activation of immune cells; it also plays an important role in breast tumorigenesis, BC progression and therapy resistance, as recently reviewed by us [205].

We have shown that particularly in luminal A BC (ER+/HER2− tumors), RANK mediates an aggressive tumor phenotype, with decreased proliferation rate and susceptibility to chemotherapy and ET [206]. Moreover, and in accordance with these observations, ectopic RANK expression was found to induce senescence in nontransformed mammary epithelia, leading to delayed, although aggressive, luminal-like tumors [207]. Stemming from this evidence, we recently reported that increased expression of RANK is associated with intrinsic and acquired resistance to CDK4/6i in preclinical HR+/HER2− BC models and that this resistance can be overcome or prevented by targeting the RANK pathway with RANKL inhibitors [136]. Analysis of patients included in the NeoPalAna clinical trial revealed not only that increased expression of RANK-metagene at baseline was predictive of resistance to CDK4/6i but also that treatment with CDK4/6i induced RANK and RANK-metagene upregulation, supporting the role of the RANK pathway in both intrinsic and acquired resistance to CDK4/6 inhibition. In our study, the transcriptomic analysis of mouse tumors highlighted chronic IFNγ response as a resistance driver, as previously suggested by another study that implicated the activation of the IFN pathway with intrinsic and acquired resistance to CDK4/6i in preclinical models and clinical HR+/HER2− BC samples [208]. Mechanistically, we propose that RANKL phosphorylates of STAT1 (Ser727), a downstream player in IFNγ signaling, independently of RANK levels. Interestingly, in our studies a STAT1 inhibitor restored the sensitivity of RANK-overexpressing BC cells to CDK4/6i, highlighting the role of the STAT1 axis in RANK-mediated resistance to CDK4/6i. However, IFN response was not the only hallmark associated with resistance in our model; other well-described mechanisms of resistance appear to be RANK-related, including CDK7 upregulation and AR and PI3K pathway activation. These promising results suggest that the pharmacological inhibition of RANKL with CDK4/6i may represent a novel therapeutic strategy in metastatic HR+/HER2− BC patients and deserve further studies.

#### 3.1.12. Autophagy

Autophagy, the lysosome-mediated degradation of molecules and subcellular elements, is extremely important for homeostasis and processes like cellular senescence, cell surface antigen presentation, energy production and protection from genome instability [209].

Upregulation of genes involved in autophagy and an increase in autophagy were previously shown to be elevated in HR+ BC cells after palbociclib treatment [98,164], while the combination of autophagy inhibitors with palbociclib-induced proliferation arrest and cause senescence in vitro and in PDXs [98].

The involvement of autophagy in chemoresistance was also observed in TNBC [210,211]. Interestingly, it was found that increased lysosomal activity was responsible for CDK4/6i resistance in CDK4/6-dependent TNBC [100]. Mechanistically, increased lysosomal activity led to CDK4/6i sequestering, and the combination of CDK4/6i with lysosomotropic or lysosome destabilizers resulted in increased sensitivity of TNBC cells to CDK4/6i. It would also be interesting to explore the possible involvement of increased lysosomal activity in CDK4/6i resistance in HR+ BC patients.

#### 3.1.13. TGF-β and EMT

The TGF-β pathway has an important role in carcinogenesis, being associated with cell proliferation, apoptosis, epithelial-to-mesenchymal transition (EMT), immune surveillance and metastasis [165,166,212]. Moreover, it was demonstrated that TGF-β contributes to G1 cell cycle arrest in BC cells [212].

It was shown that CDK4/6 inhibition can induce EMT by TGF-β signaling pathway activation [165,166,167]. Additionally, it is already established that the EMT process itself can have a pivotal role in drug resistance as well as in other malignant phenotypes [213,214].

More studies regarding the involvement of the TGF-β pathway in resistance to CDK4/6i are necessary, as is addressing the target TGF-β pathway to overcome drug resistance to CDK4/6i.

#### 3.1.14. ABC Transporters

The overexpression of adenosine triphosphate (ATP)-binding cassette (ABC) transporter proteins is well-known to be involved in drug resistance. However, despite the identification of several ABC transporters’ inhibitors, so far none gained approval [215].

The potential role of ABC transporters in resistance to CDK4/6i is not fully understood and is even contradictory. Some preclinical studies have suggested that CDK4/6i, namely palbociclib, can be a substrate for the ABCB1 and/or ABCG2 transporters, which could inclusively affect their ability to cross the blood–brain barrier [168,216] and decrease anticancer efficacy in cancer cells overexpressing the ABCB1 [169]. The upregulation of ABCB1 was also implicated in resistance to CDK7 inhibitors, reversible upon inhibition of ABC transporters [217]. However, in the opposite direction, other studies suggested that palbociclib could instead inhibit the ABCB1 transporter [218,219]. Abemaciclib was also found to reverse ABCB1- or ABCG2-mediated multidrug resistance [220]. Finally, voruciclib, one of the CDK4/6i with anticancer effects, in combination with the BRAF inhibitor vemurafenib in advanced BRAF-mutant melanoma or with the proteasome inhibitor bortezomib in TNBC xenografts was found to antagonize ABCB1- and ABCG2-mediated multidrug resistance in cancer cells [221]. Therefore, it would be important to study further what the role of ABC transporters is in response to CDK4/6i.

Overall, several mechanisms of intrinsic/acquired resistance to CDK4/6i have already been described, including alterations in cell cycle-specific and nonspecific mechanisms. Many of these are clearly nonexclusive or even overlapping. It is clear that the current knowledge on this subject is far from complete, but it allowed the identification of some targets that could be useful to treat patients that present resistance to CDK4/6i. Although not all of the aforementioned mechanisms of resistance have been successfully targeted, several novel combinations to overcome CDK4/6i resistance have been proposed. In the next section, we present a summary of the new therapeutic strategies to circumvent CDK4/6i resistance in BC that are under clinical investigation.

### 3.2. Emerging Strategies to Overcome Resistance to CDK4/6i

Currently, there are several CDK4/6i-based combination therapies being explored at the clinical level, resulting from the preclinical evidence described above. Additionally, continuing CDK4/6i is also a clinical option currently being investigated. Next, we describe the most recent results from clinical trials, also summarized in Table 4.

#### 3.2.1. Endocrine Therapy

One of the therapeutic options after progression under CDK4/6i plus ET is an ET switch. The phase II clinical trial MAINTAIN (NCT02632045) compared the effect of receiving fulvestrant or exemestane with or without ribociclib after progression on ET plus any CDK4/6i [222]. Patients treated with prior fulvestrant received exemestane and vice versa; if neither, fulvestrant or exemestane was per investigator discretion, though fulvestrant was encouraged. In the combination group, the median PFS was prolonged when compared with the monotherapy group (5.29 vs. 2.76 months), showing promising results.

Given the frequency of *ESR1* mutations following ET and/or CDK4/6i treatment, and despite their limited predictive value, several strategies to target mutant *ESR1* are starting to emerge, as recently reviewed [223].

Some of the new estrogen receptor down regulators (SERDs) already entered clinical trials. A phase I trial (NCT02734615) tested the safety and tolerability of LSZ102 as monotherapy or in combination with either ribociclib or alpelisib in HR+/HER2− BC patients who progressed under ET and demonstrated to be well tolerated alone or in combination [224]. The phase III clinical trial EMERALD (NCT03778931) compared the safety and efficacy of elacestrant versus fulvestrant or AI in HR+/HER2− BC patients who progressed under CDK4/6i plus ET, with a significant PFS improvement in patients treated with elacestrant [225]. Other new SERDs like amcenestrant (AMEERA-5, NCT04478266), camizestrant (SERENA-6, NCT04964934), giredestrant (MORPHEUS, NCT04802759), imlunestrant (EMBER-3, NCT04975308) and rintodestrant (NCT03455270) are being studied in combination with CDK4/6i in advanced HR+/HER2− metastatic BC.

Regarding new estrogen receptor modulators (SERMs), only lasofoxifene entered clinical trials in combination with CDK4/6i. The phase II ELAINEII trial (NCT04432454) is evaluating the combination of lasofoxifene with abemaciclib in advanced or metastatic HR+/HER2− BC patients with an *ESR1* mutation who had disease progression on first and/or second lines of treatment, such as ET plus CDK4/6i [226]. Promising results support the subsequent phase III trial (ELAINEIII, NCT05696626) to compare the combination of lasofoxifene plus abemaciclib versus fulvestrant plus abemaciclib.

Finally, a phase Ib/II clinical trial (NCT02448771) aimed to study the safety and effectiveness of bazedoxifene, a hybrid SERD/SERM, with palbociclib in HR+/HER2− patients who progress under other ET and did not receive prior CDK4/6i treatment. The safety profile was consistent with palbociclib monotherapy and a clinical benefit, of either a complete/partial response or stable disease, was observed in 33.3% of the patients with a PFS of 3.6 months [227].

#### 3.2.2. PI3K-AKT-mTOR Inhibitors

Alpelisib was the first specific PI3K inhibitor approved in combination with ET for advanced HR+/HER2− BC, based on the phase III trial SOLAR-1 (NCT02437318). In this trial, patients with PIK3CA-mutated BC received alpelisib plus fulvestrant or fulvestrant alone as a second-line treatment after progression on AI, with a significant improvement of the PFS in the combination group (11 months versus 5.7 months) [160]. The study included a small number of patients (5.9%) previously treated with CDK4/6i for which the median PFS favored the combination (5.5 months versus 1.8 months).

The efficacy and safety of alpelisib in combination with fulvestrant or letrozole in advanced HR+/HER- BC patients harboring *PIK3CA* mutations who progressed on or after CDK4/6i and ET therapy was then evaluated in the phase II trial BYLieve (NCT03056755). In both cohorts, alpelisib plus fulvestrant or alpelisib plus letrozole, median PFS after 6 months (primary endpoint: 7.3 months for fulvestrant [228] and 5.7 months for letrozole [229]) suggests that ET combined with alpelisib can be useful in patients that harbor PIK3CA mutations and have progressed after ET combined with CDK4/6i in first-line treatment. The phase III EPIK-B5 trial (NCT05038735) is currently studying the combination of alpelisib with fulvestrant or fulvestrant alone in patients with PIK3CA-mutated/HR+/HER2− BC who progressed under AI plus CDK4/6i.

Additionally, the TAKTIC phase Ib/II trial (NCT03959891) evaluated the efficacy of the AKT-1 inhibitor ipatasertib plus fulvestrant or AI in combination or not with palbociclib in advanced HR+/HER2− BC patients, who progressed on previous CDK4/6i treatment. The preliminary results indicate that the triple combination was well tolerated, with 8/12 patients presenting partial response or stable disease after 6 months of treatment [230]. The follow-up results of this study are awaited for more conclusions.

Another AKT inhibitor, capivasertib, was investigated in the phase III trial CAPItello-291 (NCT04305496), which included patients with HR+/HER2− advanced BC who had a relapse or disease progression during or after treatment with an AI, with (69.1%) or without previous CDK4/6i therapy [231]. In the combination group, capivasertib plus fulvestrant, PFS was 7.2 months vs. 3.6 months in the placebo–fulvestrant group (HR 0.60, 95% CI 0.51–0.71, *p* < 0.001). Therefore, AKT inhibitors may become one treatment option after CDK4/6i resistance.

Everolimus was the first mTOR inhibitor to be approved for HR+/HER2− BC based on the results from the phase III clinical trial BOLERO-2 (NCT00863655) [232]. The recent TRINITI-1 phase I/II trial (NCT02732119) assessed the efficacy of everolimus in HR+/HER2− BC patients who had previously progressed on CDK4/6i plus ET regimen. In this trial, patients receiving the triple combination of ribociclib, everolimus and exemestane (ET) had a clinical benefit at 24 weeks, with 41.1% of the patients meeting a complete/partial response or a stable disease [233]. Given this promising result, mTOR inhibitor(s) seem to be another option for patients who are resistant to CDK4/6i; and further clinical trials are expected to study the efficacy of the triple combination.

#### 3.2.3. FGFR Inhibitors

As mentioned before, the use of the FGFR pan-inhibitor erdafitinib with fulvestrant/palbociclib showed promising results in PDX models [162]. Stemming from this finding, a phase I clinical trial is addressing the safety, tolerability and efficacy of erdafitinib (NCT03238196) combined with ET and CDK4/6i in metastatic HR+/HER2− BC patients who progressed on first-line therapy with ET and CDK4/6i, with FGFR amplification. After 6 months, the median PFS was 3 months, and higher PFS (6 months) was reported in 6/8 patients with high levels of *FGFR1* amplification and in both patients with *FGFR3* amplification [234]. The clinical benefit of adding erdafitinib to the therapeutic regime will be further evaluated in a phase II trial.

#### 3.2.4. Immunotherapy

In the last years, immunotherapy emerged as a potential way to treat BC, and based on preclinical evidence, it is reasonable to hypothesize that combining CDK4/6i with immunotherapy can help to overcome resistance [79,104]. Some clinical trials are already addressing this possibility. The phase IB clinical trial JPCE (NCT02779751), investigated the efficacy and safety of abemaciclib in combination with pembrolizumab (anti-PD-1) with or without anastrozole in advanced HR+/HER2− BC patients. The combination of abemaciclib plus pembrolizumab showed antitumor activity, but high rates of adverse events were reported and the benefit/risk analysis did not support further evaluation of this combination in this BC setting [235].

On the other hand, a phase II clinical trial (NCT02778685) is investigating the triple combination of palbociclib plus fulvestrant or letrozole and pembrolizumab in newly diagnosed metastatic HR+/HER2− BC patients without prior treatment. Preliminary results indicate that the triple combination is well tolerated, and so far, a complete response rate was observed in 31% of the patients, while 25% of the patients had a partial response and 31% presented stable disease, showing some clinical benefit of the triple combination [236].

Additionally, other phase II clinical trials are ongoing to assess the efficacy of the triple combination palbociclib plus ET and avelumab (anti-PD-L1) in HR+/HER2− BC patients with early-stage (ImmunoADAPT, NCT03573648) [237] or advanced (PACE, NCT03147287) [238,239] BC disease. Preliminary results from the PACE trial showed encouraging results, with an increase in PFS of 8.1 months in the group receiving the triple combination compared to 4.8 and 4.6 months in the groups receiving fulvestrant alone and palbociclib plus fulvestrant, respectively [240].

#### 3.2.5. Chemotherapy

Although CDK4/6i are not expected to synergize with chemotherapy, given their cytostatic effect, the use of non-cell-cycle-specific chemotherapy in combination with CDK4/6i to enhance their efficacy is starting to be studied. One phase I clinical trial (NCT01320592) evaluated the combination of palbociclib with paclitaxel in Rb-expressing metastatic BC patients, showing that the combination is safe, and future studies are needed to compare the PFS and response rates [241].

It is also hypothesized that chemotherapy after acquired resistance to ET plus CDK4/6i can be effective [242], and in clinical practice is frequently chosen as the follow-up treatment plan. Currently, the clinical trials evaluating the clinical efficacy of chemotherapy as a monotherapy or in combination with other compounds in patients with ER+/HER2 BC that presented resistance to CDK4/6i and ET are the phase III TROPiCS-02 (NCT03901339), the phase III KEYNOTE-B49 (NCT04895358), the phase II TATEN (NCT04251169) and a phase I trial (NCT04134884).

#### 3.2.6. CDK7 Inhibitors

The efficacy of selective CDK7 inhibitors, namely, CT-7001 (NCT03363893) and SY 5609 (NCT04247126), alone or in combination with other therapies in patients with advanced solid tumors, including advanced HR+/HER2− BC patients that progressed on CDK4/6i plus ET, is currently being investigated, as reviewed by [243]. Given the preclinical data supporting the prominent role of CDK7-mediated resistance to CDK4/6i, results are expected with enthusiasm.

#### 3.2.7. BCL2 Inhibitors

The selective BCL2 inhibitor ABT-199 (venetoclax) was primarily approved for acute myeloid leukemia and chronic lymphocytic leukemia patients [244]. However, a significant efficacy of venetoclax in combination with tamoxifen was observed in HR+/HER2− BC patients with BCL2 overexpression (ISRCTN98335443) [245]. The ongoing phase I clinical trial PALVEN (NCT03900884) is evaluating the safety and efficacy of AI plus CDK4/6i combined with venetoclax as a first-line treatment in metastatic HR+/HER2− BC patients with BCL2 overexpression.

Additionally, the phase II clinical trial VERONICA (NCT03584009), terminated at the end of 2022, assessed the efficacy of venetoclax plus fulvestrant compared with fulvestrant alone in patients who progressed under CDK4/6i therapy. However, this study reported no significant difference in PFS between the combination and monotherapy arms (2.69 months versus 1.94 months) [246]. OS results are awaited.

#### 3.2.8. Aurora Kinase A Inhibitors

Currently, the Aurora kinase A selective inhibitor erbumine (LY3295668) is being tested in a phase I clinical trial (NCT03955939) as a monotherapy or combined with ET in HR+/HER2− BC patients previously treated with CDK4/6i. Other mitotic kinase inhibitors were tested in preclinical models, showing good results that might encourage future clinical trials [187].

#### 3.2.9. Histone Deacetylase (HDAC) Inhibitors

HDAC inhibitors (HDACi) are known to have potent antitumor effects, inducing cell death, cell cycle arrest, senescence and autophagy, amongst other effects [247]. Tucidinostat (or chidamide) is an oral subtype-selective HDACi with antitumor activity reported in an exploratory study in patients with advanced HR+ BC, in combination with exemestane [248]. Later, the ACE phase III trial (NCT02482753) showed that tucidinostat plus exemestane improved PFS vs. placebo and exemestane in patients with advanced HR+/HER2− BC that progressed after previous ET [249]. More recently, another study analyzed the efficacy and safety of tucidinostat combined with ET in patients after prior CDK4/6i progression [250]. At a median follow-up of 10 months, the median PFS was 2.0 months (95% CI 1.9–2.1), and the median OS was 14 months (95% CI 6.3–21.7), suggesting that tucidinostat plus ET may be an optional sequential strategy for patients with HR+/HER2− advanced BC that has progressed on CDK4/6i.

**Table 4 cancers-15-04835-t004:** Recently completed and ongoing clinical trials for metastatic HR+/HER2− BC after progression under ET and/or CDK4/6i.

Therapy	Clinical Setting	Phase	Intervention	Trial Identifier	Status *	Results
Endocrine therapy	Disease progression under ET+ CDK4/6i.	II	Fulvestrant or exemestane vs. fulvestrant or exemestane + ribociclib	MAINTAIN (NCT02632045)	Completed	Fulvestrant or exemestane + ribociclib increased PFS compared to ET alone [223].
	Disease progression under ET.	Ib/II	Bazedoxifene + palbociclib	NCT02448771	Completed	Clinical benefit, complete/partial response or stable disease were observed [227].
	Disease progression under ET.	I	LSZ102 vs. LSZ102 + ribociclib vs. LSZ102 + alpelisib	NCT02734615	Completed	LSZ102 well tolerated alone or in combination. Preliminary clinical activity observed in combination groups [224].
	Disease progression under ET + CDK4/6i with *ESR1* mutations.	III	Elacestrant vs. fulvestrant or AI	EMERALD (NCT03778931)	Active, not recruiting	PFS improvement in elacestrant arm [225].
	Disease progression under ET.	I	Rintodestrant vs. rintodestrant + palbociclib	NCT03455270	Active, not recruiting	Rintodestrant demonstrated a safety/tolerability profile as monotherapy or combined with palbociclib [251].
	Disease progression under ET + CDK4/6i.	III	ET vs. imlunestrant vs. imlunestrant + abemaciclib	EMBER-3 (NCT04975308)	Recruiting	N/A
	Disease progression under AI + Palbociclib or Abemaciclib with *ESR1* Mutations.	III	Camizestrant + palbociclib or abemaciclib vs. AI + palbociclib or abemaciclib	SERENA-6 (NCT04964934)	Recruiting	N/A
	Disease progression under ET + CDK4/6i.	Ib/II	Giredestrant vs. giredestrant + abemaciclib or ipatasertib or inavolisib or ribociclib or everolimus or samuraciclib or atezolizumab or abemaciclib + atezolizumab	MORPHEUS (NCT04802759)	Recruiting	N/A
	Disease progression under ET + CDK4/6i.	III	Giredestrant + everolimus vs. everolimus + exemestane	NCT05306340	Recruiting	NA
	Disease progression under AI + palbociclib or ribociclib with *ESR1* mutations.	III	Lasofoxifene + abemaciclib vs. fulvestrant + abemacicclib	ELAINE III (NCT05696626)	Not yet recruiting	N/A
PI3K-AKT-mTOR inhibitors	Disease progression under ET or ET + CDK4/6i.	I/II	Exemestane + ribociclib + everolimus	TRINITI-1 (NCT02732119)	Completed	Triple combination with clinical benefit at 24 weeks [233].
	Disease progression under AI.	III	Fulvestrant vs. fulvestrant + capivasertib	CAPItello-291 (NCT04305496)	Active, not recruiting	Capivasertib + fulvestrant increased PFS compared with fulvestrant alone [231].
	Disease progression under AI + CDK4/6i with *PIK3CA* mutations.	II	Alpelisib + fulvestrant or letrozole	BYLieve (NCT03056755)	Active, not recruiting	Alpelisib demonstrated clinical activity in combination with fulvestrant or letrozole [252].
	Disease progression under ET or ET+ CDK4/6i.	Ib/II	Ipatasertib + fulvestrant or letrozole vs. ipatasertib + fulvestrant or letrozole + palbociclib	TAKTIC (NCT03959891)	Active, not recruiting	Triple combination well tolerated. Partial response or stable disease observed [230].
	Disease progression under AI + CDK4/6i with *PIK3CA* Mutations.	III	Fulvestrant vs. alpelisib + fulvestrant	EPIK-B5 (NCT05038735)	Recruiting	N/A
	Disease progression under ET + CDK4/6i.	I	Inavolisib + letrozole/fulvestrant	NCT03006172	Recruiting	NA
	Disease progression under AI + CDK4/6i.	III	Fulvestrant vs. ipatasertib + fulvestrant	NCT04650581	Recruiting	NA
FGFR inhibitors	Disease progression under ET + CDK4/6i with *FGFR* amplification.	Ib	Erdafitinib + fulvestrant + palbociclib	NCT03238196	Active, not recruiting	Increased PFS in patients with high levels of *FGFR1* amplification [234].
Immunotherapy	Disease progression under AI + CDK4/6i.	II	Fulvestrant vs. fulvestrant + palbociclib vs. fulvestrant + palbociclib + avelumab	PACE (NCT03147287)	Active, not recruiting	Increased PFS in the triple-combination group [240].
	Disease progression under ET + CDK4/6i with PD-L1 Expression.	III	CT vs. CT + pembrolizumab	KEYNOTE-B49 (NCT04895358)	Recruiting	N/A
Chemotherapy	Disease progression under ET + CDK4/6i.	II	Pembrolizumab + paclitaxel	TATEN (NCT04251169)	Active, not recruiting	N/A
	Disease progression under ET + CDK4/6i.	I	ASTX727 + talazoparib	NCT04134884	Recruiting	N/A
	Disease progression under ET + CDK4/6i.	I/II	CT7001 + fulvestrant	NCT03363893	Completed	Tolerable safety [253].
CDK7 inhibitors	Disease progression under ET + CDK4/6i.	I	SY-5609 + fulvestrant	NCT04247126	Active, not recruiting	N/A
	Disease progression under CDK4/6i.	II	Fulvestrant vs. fulventrant + venetoclax	VERONICA (NCT03584009)	Completed	No significant difference in PFS [246].
BCL2 inhibitors	Disease progression under ET with *BCL2* expression.	Ib	Letrozole + palbociclib + venetoclax	PALVEN (NCT03900884)	Recruiting	N/A
	Disease progression under ET + CDK4/6i.	Ib	Erbumine vs. erbumine + ET	NCT03955939	Completed	N/A
Aurora kinase A inhibitors	Patients pretreated with CDK4/6i.	I	Xentuzumab + abemaciclib	NCT03099174	Active, not recruiting	NA
IGF inhibitors	Disease progression under ET + CDK4/6i.	II	PF-06873600 vs. PF-06873600 + ET	NCT03519178	PF-06873600 vs. PF-06873600 + ET	N/A
CDK4/6 inhibitors	Disease progression under AI or tamoxifen ± LHRHa + CDK4/6i.	II	Palbociclib + fulvestrant	NCT04318223	Palbociclib + fulvestrant	N/A
	Clinical benefit of 1st line palbociclib + ET.	II	Palbociclib rechallenge + ET	PALMIRA (NCT03809988)	Clinical benefit of 1st line Palbociclib + ET	N/A
	Disease progression under ET + CDK4/6i.	III	Abemaciclib + fulvestrant	Post-MONARCH (NCT05169567)	Recruiting	NA
ADC	Disease progression under ET + CDK4/6i and CT.	III	Sacituzumab govitecan vs. CT	TROPiCS-02 (NCT03901339)	Active, not recruiting	Increased OS in the sacituzumab govitecan group [254].
	Chemotherapy naïve disease progression under ET with HR+/Her-2 low or ultralow.	III	T-DXD vs. investigator choice chemotherapy	DB-06 (NCT04494425)	Recruiting	NA
	Endocrine resistant disease.	III	Dato-Dxd	TROPION Breast01 (NCT04494425)	Recruiting	NA

ADC: antibody-drug conjugates; AI: aromatase inhibitor; CT: chemotherapy; ET: endocrine therapy; LHRHa: luteinizing hormone-releasing hormone (LHRH) agonists; N/A: not available; PFS: progression-free survival; OS: overall survival. * At the time of submission of this work.

## 4. Conclusions and Future Perspectives

After several years of research, the development of selective, potent CDK4/6i is a major success in HR+/HER2− BC therapy, and combination with ET became the first-line treatment choice in advanced disease. Given the promising results from ongoing clinical trials, it is expected that, in the near future, the use of CDK4/6i will expand to patients with early BC and with other BC subtypes. Moreover, although CDK4/6 inhibition has been proven effective in other solid tumors, the translation for its use in clinical practice is still awaited, and intensive investigation is still ongoing to support the clinical efficacy of CDK4/6i beyond BC.

Since the discovery of CDK4/6i, the understanding of their mechanisms of action has advanced significantly, although they are still not completely understood. It is now clear that the inhibition of CDK4/6 affects tumor cells beyond cell cycle arrest and, more importantly, may affect nontumor cells, being very likely that these compounds may have a broader effect than currently appreciated. Hence, further studies on this topic are extremely important to leverage the utility of CDK4/6i. A good example is the currently ongoing clinical trials testing CDK4/6i in combination with immunotherapy, empowered by the observations that CDK4/6i enhance antitumor immune response.

On the other hand, and despite the great efficacy of CDK4/6i in the approved clinical settings, resistance to these inhibitors seems unavoidable. Approximately 10–20% of patients are intrinsically resistant, and most will almost certainly develop acquired resistance in the course of treatment. Numerous studies have investigated the alterations behind escape mechanisms to CDK4/6 inhibition, and multiple mechanisms of resistance have been described, either cell cycle-specific or nonspecific. This investigation has been the basis for ongoing clinical trials testing new therapeutic strategies that may overcome or delay resistance to CDK4/6i. However, notwithstanding the number of published studies and the current knowledge about the molecular mechanisms of resistance to CDK4/6i, a complete and detailed picture with clinical validation is still missing. An important factor to consider is that CDK4/6i are used in combination with ET, and the majority of preclinical studies are based on CDK4/6i monotherapy. Therefore, it is essential to investigate how ET and CDK4/6i interact and cooperate for therapy resistance.

Finally, the identification of well-established biomarker(s) in select patients that are more likely to present a favorable response to CDK4/6i treatment would be a major achievement. To date, several studies tackled this question, unfortunately without clinical translation. Thus, further studies are required to respond to this unmet medical need.

In summary, several unresolved questions maintain the research about CDK4/6i as an active topic. Overall, there are three pressing medical needs in the CDK4/6i field: (1) the identification and clinical validation of predictive biomarker(s), (2) a deeper clarification of the mechanisms involved in CDK4/6i plus ET treatment failure, and, finally, (3) the establishment of novel treatment combinations to prevent/overcome resistance to CDK4/6i plus ET.

## Figures and Tables

**Figure 1 cancers-15-04835-f001:**
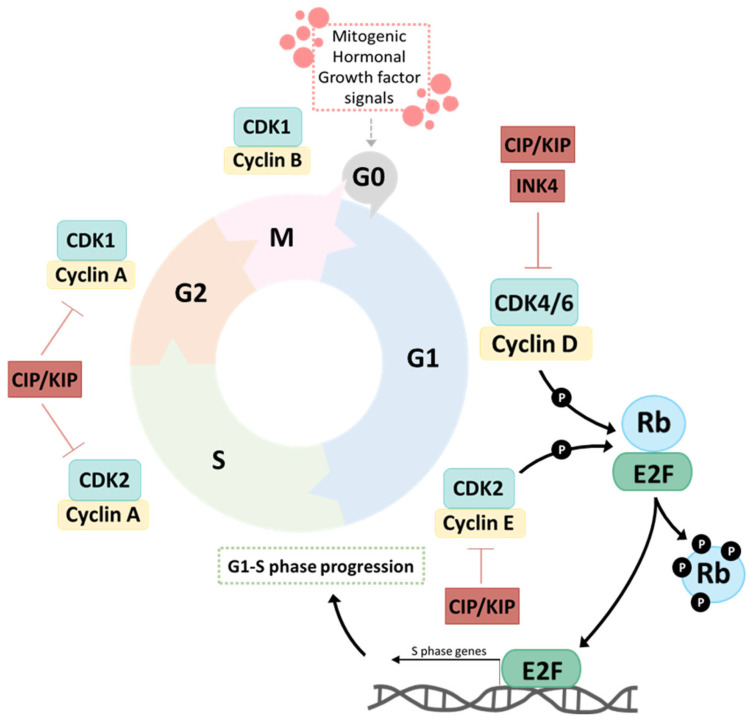
Schematic representation of the eukaryotic cell cycle. Different mitogenic, hormonal and growth factor signals can trigger eukaryotic cells to enter in the cell cycle. The entry in the cell cycle is mainly controlled by the CDK4/6 proteins that bind to cyclin D proteins, forming active cyclin D–CDK4/6 complexes. These active complexes phosphorylate Rb protein, allowing the release of E2F transcription factor. The free E2F further stimulate *CCNE* (cyclin E) expression, which binds to CDK2 and forms the cyclin E–CDK2 complex that leads to Rb hyperphosphorylation, promoting cell cycle transition to S phase. Cyclin A and cyclin B form complexes with CDK2 and CDK1, promoting S/G2, G2/M transition and ultimately the entry in mitosis (M phase). To prevent inappropriate cell division, cell cycle progression can be suppressed by two families of CDK inhibitors, INK4 that comprises p16INK4A, p15INK4B, p18INK4C and p19INK4D proteins and CIP/KIP that includes p21Waf1/Cip1, p27Kip1 and p57Kip2 proteins.

**Figure 2 cancers-15-04835-f002:**
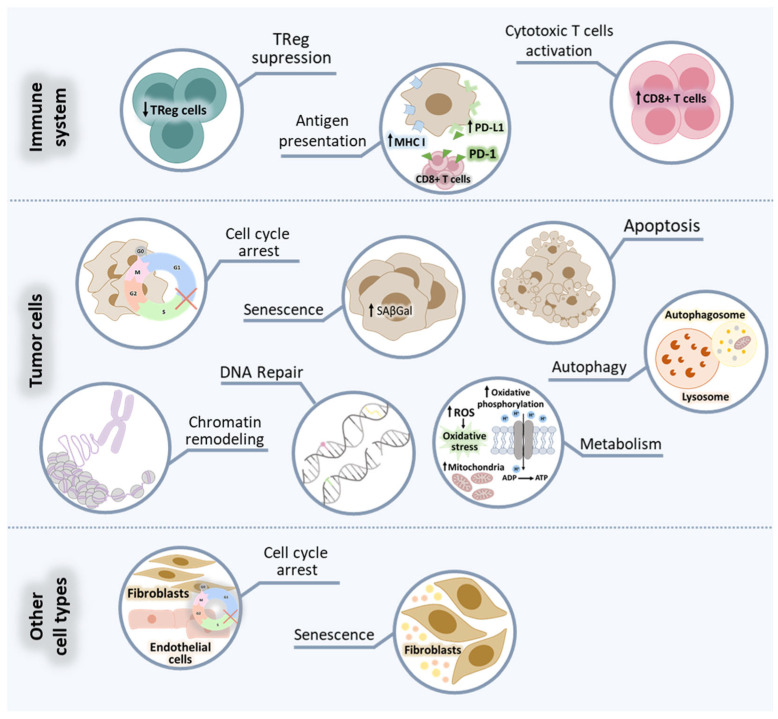
Cellular processes affected by CDK4/6i. ADP: adenosine diphosphate; ATP: adenosine triphosphate; MHC I: major histocompatibility complex class I; PD-1: programmed cell death protein 1; PD-L1: PD-ligand 1; ROS: reactive oxygen species; SaβGal: senescence-associated β-galactosidase; TReg: regulatory T cells.

**Table 1 cancers-15-04835-t001:** CDK4/6i approved or in clinical trials to treat breast cancer.

Compound	Company	Clinical Status	Selectivity (IC_50_ or K_i_)
Palbociclib (PD0332991)	Pfizer	Approved. HR+/HER2− advanced or metastatic BC in combination with ET.	CDK4: 11 nM (IC^50^) CDK6: 16 nM (IC^50^)
Ribociclib (LEE011)	Novartis	Approved. HR+/HER2− advanced or metastatic BC.	CDK4: 10 nM (IC_50_) CDK6: 39 nM (IC_50_)
Abemaciclib (LY2835219)	Eli Lilly	Approved. HR+/HER2− advanced or metastatic BC in combination with ET. HR+/HER2− advanced or metastatic BC as monotherapy. Adjuvant therapy for high-risk, early-stage HR+/HER2− BC in combination with ET.	CDK4: 2 nM (IC_50_) CDK6: 10 nM (IC_50_) CDK9: 57 nM (IC_50_)
Dalpiciclib (SHR6390)	Jiangsu Hengrui Medicine	In clinical trials. Phase III for HR+/HER2− BC in combination with ET. Phase I/II for multiple tumor types in combination with ET or immunotherapy.	CDK4: 12 nM (IC50) CDK6: 10 nM (IC50)
PF-06873600	Pfizer	In clinical trials. Phase II for metastatic HR+/HER2− and TNBC.	CDK2: 0.09 nM (K_i_) CDK4: 0.13 nM (K_i_) CDK6: 0.16 nM (K_i_)
Trilaciclib (G1T28)	G1 Therapeutics	In clinical trials. Phase III for early-stage and metastatic TNBC in combination with chemotherapy.	CDK4: 1 nM (IC_50_) CDK6: 4 nM (IC_50_) CDK9: 50 nM (IC_50_)
Lerociclib (G1T38)	G1 Therapeutics	In clinical trials. Phase I/II for HR+/HER2− metastatic BC in combination with ET.	CDK4: 1 nM (IC_50_) CDK6: 2 nM (IC_50_) CDK9: 28 nM (IC_50_)
R547	Hoffmann-La Roche	In clinical trials. Phase I for advanced BC and other solid cancers.	CDK1: 2 nM (K_i_) CDK2: 3 nM (K_i_) CDK4: 1 nM (K_i_)

BC: breast cancer; ET: endocrine therapy; HER2: human epidermal growth factor receptor 2; HR: hormone receptor; TNBC: triple-negative breast cancer.

**Table 2 cancers-15-04835-t002:** Clinical trials evaluating CDK4/6i efficacy in early-stage HR+/HER2− breast cancer.

Neoadjuvant Studies					
Phase	BC Stage	Intervention	Trial Identifier	Status *	Results
III	Stage II–III	Palbociclib + ET vs. ET	SAFIA (NCT03447132)	Completed	No statistically significant differences in pCR rates [55].
II	Stage I–III	AI vs. AI then AI + palbociclib vs. palbociclib then AI + palbociclib vs. AI + palbociclib	PALLET (NCT02296801)	Completed	Palbociclib + AI (all arms) significantly decreased Ki67 compared to AI alone but did not increase the pCR rate [44].
II	Stage I–III	Ribociclib + AI vs. chemotherapy	CORALLEEN (NCT03248427)	Completed	No significant differences in ROR. Ribociclib + AI was associated with better HRQoL outcomes [56].
II	Stage I–III	AI vs. AI + abemaciclib vs. abemaciclib then AI + abemaciclib	neoMONARCH (NCT02441946)	Completed	AI + abemaciclib or abemaciclib alone significantly decreases Ki67 compared to AI alone [57].
II	Stage II–III	AI vs. ribociclib + AI vs. ET	FELINE (NCT02712723)	Active, not recruiting	PEPI score was equal in AI and ribociclib + AI groups [58].
II	Stage II–III	AI vs. AI then palbociclib + AI	NeoPalAna (NCT01723774)	Active, not recruiting	The CCCA rate was significantly higher after adding palbociclib to AI [59].
II	Stage II–III	AI vs. AI then chemotherapy vs. AI + ribociclib	NEOBLC (NCT03283384)	Active, not recruiting	N/A
II	Stage II	Ribociclib + AI	RIBOLARIS (NCT05296746)	Recruiting	N/A
**Adjuvant Studies**					
**Phase**	**BC Stage**	**Intervention**	**Trial Identifier**	**Status ***	**Results**
III	High risk of recurrence	Palbociclib + ET vs. ET	PENELOPE-B (NCT01864746)	Active, not recruiting	Palbociclib in addition to ET did not improve iDFS [60].
III	Stage II–III	Palbociclib + ET vs. ET	PALLAS (NCT02513394)	Active, not recruiting	Palbociclib in addition to ET did not improve iDFS [61].
III	High risk of recurrence	Abemaciclib + ET vs. ET	MonarchE (NCT03155997)	Active, not recruiting	Abemaciclib + ET reduces significantly ROR compared to ET alone [45].
III	Stage II–III	Ribociclib + ET vs. ET	NATALEE (NCT03701334)	Active, not recruiting	Ribociclib + ET improves iDFS compared to ET alone [62].
III	High risk of recurrence	Abemaciclib + ET vs. ET	ADAPTlate (NCT04565054)	Recruiting	N/A
III	Intermediate risk of recurrence	Ribociclib + ET vs. chemotherapy	ADAPTcycle (NCT04055493)	Recruiting	N/A
II	High risk of recurrence	Palbociclib + ET vs. ET	HIPEx (NCT04247633)	Recruiting	N/A
II	stage II–III	Ribociclib + ET vs. ET	LEADER (NCT03285412)	Recruiting	N/A

AI: aromatase inhibitor; CCCA: complete cell cycle arrest; ET: endocrine therapy; HRQoL: health-related quality of life; iDFS: invasive disease-free survival; N/A: not available; pCR: pathological clinical response; PEPI: preoperative endocrine prognostic index; ROR: risk of relapse. * At the time of submission of this work.

**Table 3 cancers-15-04835-t003:** Major mechanisms implicated in intrinsic/acquired resistance to CDK4/6i.

Mechanism/Player	Impact on Response to CDK4/6i	Clinical Potential
Protein Rb	Impaired Rb function abrogates response [29,118,119,120,121,122,123].	Prognostic/predictive role not validated [124,125]
Cyclin D–CDK4–CDK6	Upregulation of cyclin D [118,126,127], CDK4 [127,128,129] and/or CDK6 [118,129,130,131,132,133] associated with resistance.	Prognostic/predictive role not validated [18,124].
Cyclin E–CDK2	Upregulation ofcCyclin E1, cyclin E2 and CDK2 associated with resistance [59,118,120,124,129,134,135].	Prognostic/predictive role not validated [50,125].
CDK7	Upregulation of CDK7 associated with resistance [128,136].	Not assessed.
INK4 and CIP/KIP members	p16 overexpression in Rb-proficient models [123,137] and increased phosphorylation of p27 [120,138] associated with decreased sensitivity.	Prognostic/predictive role not validated [124,125].
Other cell cycle regulators	Overexpression of WEE1 or MDM2 associated with intrinsic resistance [139,140].	Combination with MDM2 inhibitor could abrogate resistance to CDK4/6i and ET in preclinical models [141].
	FZR1 KD associated with intrinsic resistance [142].	Not assessed.
	AP-1 and c-Fos increased upon acquired resistance [143].	AP-1 blockade combined with palbociclib could effectively inhibit cell proliferation [143].
	TK1 overexpression associated with intrinsic resistance [144].	Potential prognostic value of plasma TKa reported (TREnd, NCT02549430) [145] and under investigation (BioItaLEE, NCT03439046; PYTHIA, NCT02536742).
	Amplification of *AURKA* (Aurora kinase A) found in tumor biopsies from resistant patients [134].	Not assessed.
	Mutated *MYC* was found in patients treated with abemaciclib plus AI [146] c-MYC induction and cyclin E/CDK2 activity followed CDK4/6i therapy [147].	Not assessed.
	Activated c-MET found in patients treated with abemaciclib plus AI [146].	Not assessed.
HRs and HER2	Loss of ER/PR expression [128,129] and activation of AR signaling [148] associated with intrinsic and acquired resistance. HER2 mutations conferred estrogen independence as well as resistance to ET and CDK4/6i [149]. *ERBB2* amplification in patients treated with CDK4/6i [134,150].	Prognostic/predictive role of *ESR1* mutations not validated [151,152]. Combination of palbociclib with enzalutamide, a selective AR inhibitor, abrogated resistance in BC cell lines [148]. Anti-HER2 neratinib abrogated resistance to ET+CDK4/6i [149].
PI3K-AKT-mTOR pathway	Activation of PI3K-AKT-mTOR signaling associated with intrinsic and acquired resistance [118,126,134,153,154,155,156,157,158,159].	Combination of PI3K-AKT-mTORi and CDK4/6i has been proven to overcome/prevent/delay intrinsic and acquired resistance [118,126,155,156,159,160]. *PIK3CA* mutations in plasma ctDNA [151,161] or increased levels of activated AKT [154] may have predictive potential.
FGFR pathway	FGFR1 upregulation associated with acquired resistance [50,128,152,162].	Combination of CDK4/6i with FGFR inhibitor could abrogate acquired resistance [162]. *FGFR1* amplification (MONALEESA-2 and PALOMA-3) [50,152,162] or *FGFR2* mutations [134] may have prognostic potential.
MAPK-ERK pathway	Activation of MAPK-ERK signaling associated with intrinsic and acquired resistance [128,163].	Combination of MEK inhibitors with CDK4/6i plus ET was shown to be effective in blocking cells proliferation [162,163].
RANK-RANKL pathway	RANK OE associated with intrinsic resistance; RANK upregulation observed upon acquired resistance [136].	Combination of CDK4/6i with RANKL inhibitors could abrogate/delay intrinsic and acquired resistance [136].
Autophagy	Upregulation of genes involved in autophagy and an increase in autophagy observed upon treatment [98,164]. Increased lysosomal activity associated with resistance in TNBC [100].	Combination of autophagy inhibitors with palbociclib-induced proliferation arrest and senescence in preclinical models [98]. Combination of CDK4/6i with lysosomotropic or lysosome destabilizers resulted in increased sensitivity of TNBC cells to CDK4/6i [100].
TGF-β and EMT	CDK4/6i could induce EMT by TGF-β signaling pathway activation [165,166,167].	Not assessed.
ABC transporters	Overexpression of ABCB1 and/or ABCG2 transporters may decrease anticancer efficacy of palbociclib [168,169].	Not assessed.

ctDNA: circulating tumor DNA; ET: endocrine therapy; HRs: hormone receptors; KD: knockdown; OE: overexpression; TKa: thymidine kinase 1 (TK) activity; TNBC: triple-negative breast cancer.

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
