# Peer review of "The Evolving Pathways of the Efficacy of and Resistance to CDK4/6 Inhibitors in Breast Cancer"

_cancers, 2023, doi:10.3390/cancers15194835_

Round 1

Reviewer 1 Report

The paper entitled “The evolving pathways of efficacy and resistance to CDK4/6 inhibitors in breast cancer” describes the topic of a novel and relevant group of anti-cancer drugs - Cyclin-Dependent Kinase inhibitors. The authors provide in-depth insight into the topic. Despite the article being well-written, and by any means I recommend it for publication in the Journal, I have a number of suggestions for the authors:

  1. In sections 2.1.1; 2.1.2; and 2.1.3; the authors should consider providing the number of patients included in a particular study and p values related to the described results
  2. In section 2.2 the authors describe the mechanism of actions of CDK4/6 inhibitors. In lines 260 and 261, the authors mentioned that the drugs may affect E2F-dependent processes such as DNA repair or apoptosis. However, in Figure 2, the authors did not show either DNA repair or apoptosis as a result of CDK4/6 inhibition. I believe that both cellular processes are crucial in terms of anti-cancer therapy. Please consider mentioning those processes in Figure 2, if possible.
  3. Also, I believe that adding a section describing potential ways to counter the CDKi resistance due to the activity of ABC family transporters would be beneficial for the paper. 
  4. Please consider adding a paragraph showing the limitations (other than CDKi resistance) of CDKi use in either the Introduction or Conclusion section.

Author Response

Reviewer #1: The paper entitled “The evolving pathways of efficacy and resistance to CDK4/6 inhibitors in breast cancer” describes the topic of a novel and relevant group of anti-cancer drugs - Cyclin-Dependent Kinase inhibitors. The authors provide in-depth insight into the topic. Despite the article being well-written, and by any means I recommend it for publication in the Journal, I have a number of suggestions for the authors.

The authors’ acknowledge the thorough revision of the manuscript, and the extremely positive feedback from the reviewer.

  1. In sections 2.1.1; 2.1.2; and 2.1.3; the authors should consider providing the number of patients included in a particular study and p values related to the described results.

The authors appreciate this important suggestion. In the revised manuscript, trials described in sections 2.1.1, 2.1.2, and 2.1.3 include now the mention to number of enrolled patients, and results include HR, 95%CI and p-value.

  1. In section 2.2 the authors describe the mechanism of actions of CDK4/6 inhibitors. In lines 260 and 261, the authors mentioned that the drugs may affect E2F-dependent processes such as DNA repair or apoptosis. However, in Figure 2, the authors did not show either DNA repair or apoptosis as a result of CDK4/6 inhibition. I believe that both cellular processes are crucial in terms of anti-cancer therapy. Please consider mentioning those processes in Figure 2, if possible.

The authors appreciate this excellent suggestion. In the revised manuscript Figure 2 includes reference to DNA repair and apoptosis.

  1. Also, I believe that adding a section describing potential ways to counter the CDKi resistance due to the activity of ABC family transporters would be beneficial for the paper. 

We acknowledge this suggestion. It is true that no revision can depict the entire complexity of resistance to CDK4/6i. Nevertheless, in the revised manuscript, section 3.1 (and Table 3) includes relevant evidence regarding the role of ABC transporters in resistance to CDK4/6i.

  1. Please consider adding a paragraph showing the limitations (other than CDKi resistance) of CDKi use in either the Introduction or Conclusion section.

The authors appreciate the suggestion. We believe that throughout the review and reinforced in the Conclusions we address the major limitation of CDK4/6i beyond resistance, namely the lack of predictive biomarkers.

Reviewer 2 Report

I am please to have the opportunity to read and evaluate this review. It is generally well written and and covers a topic relevant to daily clinical practice.

I would suggest adding other mechanisms of resistance such as regulation by microRNAs , cMet and cMyc aberrations , KRAS/HRAS/NRAS mutations/amplification, Human epidermal growth factor receptor 2 (HER2) mutations.

Also I suggest in the initial part to add a paragraph to explain the difference between endocrine resistance and resistance to CDK4/6 inhibitors.

Finally, I suggest adding a final table with current combination studies. 

Author Response

Reviewer #2: I am pleased to have the opportunity to read and evaluate this review. It is generally well-written and covers a topic relevant to daily clinical practice.

  1. I would suggest adding other mechanisms of resistance such as regulation by microRNAs, cMet and cMyc aberrations, KRAS/HRAS/NRAS mutations/amplification, Human epidermal growth factor receptor 2 (HER2) mutations.

We acknowledge this suggestion. It is true that no revision can depict the entire complexity of resistance to CDK4/6i. Nevertheless, in the revised manuscript, section 3.1 (and Table 3) includes relevant evidence regarding the role of cMet and cMyc aberrations, additional data on KRAS/HRAS/NRAS mutations/amplification, which was already referred, and HER2 mutations in resistance to CDK4/6i. Regarding miRNAs, we have mentioned evidence whenever an association between a miRNA and another mechanism of resistance was observed.

  1. Also, I suggest in the initial part to add a paragraph to explain the difference between endocrine resistance and resistance to CDK4/6 inhibitors.

We do appreciate this very good suggestion, which we incorporated in the Introduction section.

  1. Finally, I suggest adding a final table with current combination studies. 

We appreciate the suggestion and included Table 4 in the revised manuscript, which summarizes the current studies for metastatic HR+/HER2- BC after progression under ET and/or CDK4/6i.

Reviewer 3 Report

Very well written and comprehensive review into the various CDK 4/6 inhibitors, their indications as well as resistance mechanisms and potential strategies to overcome them. 

This research addresses the role of various CDK 4 6 inhibitors in management of breast cancer.  It also touches upon the role of CDK 4/6 inhibitors potentially in other malignancies where they have been studied.  The authors go into the details of various proposed resistance mechanisms to CDK 4/6 inhibitors and clinical trials exploring potential options for overcoming such resistance.

This topic is original and very relevant in this field since CDK 4/6 inhibitors are the cornerstone of management of metastatic hormone receptor positive and HER2 negative breast cancer.  This manuscript does address the current gaps in the field which relate to identifying potential resistance mechanisms to CDK 4/6 inhibitors and potentially looking at the options that could be relevant to overcoming such resistance.

This manuscript summarizes the various resistance mechanisms in a comprehensive very, goes into details of how CDK 4/6 inhibitors got approved in management of hormone receptor positive and HER2 negative breast cancer as well as comprehensively goes into details of which potential drugs code target the various resistance mechanisms to CDK 4/6 inhibitors.

The conclusions are consistent with the evidence and arguments presented and they address the main question posed. The references are appropriate. The tables and figures are very comprehensive and appropriate.

Author Response

The authors’ acknowledge the thorough revision of the manuscript, and the extremely positive feedback from the reviewer.

Reviewer 4 Report

The current article is a well written, informative, and comprehensive review, providing valuable information on the proposed mechanisms of action of CDK4/6 inhibitors. The review also focuses on the potential mechanisms of breast cancer resistance to CDK4/6 inhibitors, which is very important considering the unmet need for reliable predictive biomarkers. In addition, it presents valuable data on previous and ongoing therapeutic combination strategies, including numerous pre-clinical and clinical trials, and providing the rational for these combination strategies. Overall, the article structure is very good, and I believe will satisfy the readers.

Major comment

I would suggest the authors to discuss the potential mechanisms and biomarkers of resistance, discriminating, if possible, between intrinsic and acquired resistance.

Minor comments

Comment 1

(3.2. Emerging strategies to overcome resistance to CDK4/6i) :

Another targeted drug useful to mention is Histone Deacetylase Inhibitor tucidinostat, tested in the ACE phase III trial in postmenopausal women with HR+advanced breast cancer resistant to endocrine therapy, randomly assigned to receive exemestane with tucidinostat or placebo, that showed longer PFS in the tucidinostat group. 

Comment 2

(3.2.2. PI3K-AKT-mTOR inhibitors):

It may be useful to mention CAPItello-291  phase III study evaluating the addition of AKT-inhibitor capivasertib to fulvestrant that showed a significant improvement in PFS in patients who had progressed to previous AI with or without CDK4/6 inhibitor (in February 2023, the FDA granted priority review designation to cabivasertib + fulvestrant) 

Needs linguistic improvement

Author Response

Reviewer #4: The current article is a well-written, informative, and comprehensive review, providing valuable information on the proposed mechanisms of action of CDK4/6 inhibitors. The review also focuses on the potential mechanisms of breast cancer resistance to CDK4/6 inhibitors, which is very important considering the unmet need for reliable predictive biomarkers. In addition, it presents valuable data on previous and ongoing therapeutic combination strategies, including numerous pre-clinical and clinical trials, and providing the rational for these combination strategies. Overall, the article structure is very good, and I believe will satisfy the readers.

The authors’ acknowledge the thorough revision of the manuscript, and the extremely positive feedback from the reviewer.

  1. Major comment - I would suggest the authors to discuss the potential mechanisms and biomarkers of resistance, discriminating, if possible, between intrinsic and acquired resistance.

We do appreciate this important comment, and totally agree that most of the resistance mechanisms known so far can be related with both intrinsic and acquired resistance and it is particularly difficult to clearly implicate the cellular aspects in one or another. We added this remark in the Introduction, as well as in the beginning of section 3 and title of Table 3. Whenever possible to distinguish between intrinsic or acquired resistance this was clarified throughout the text.

  1. Minor comments

2.1. Comment 1 (3.2. Emerging strategies to overcome resistance to CDK4/6i): Another targeted drug useful to mention is Histone Deacetylase Inhibitor tucidinostat, tested in the ACE phase III trial in postmenopausal women with HR+advanced breast cancer resistant to endocrine , randomly assigned to receive exemestane with tucidinostat or placebo, that showed longer PFS in the tucidinostat group. 

We do appreciate the excellent suggestion. In fact, there is a study reporting that tucidinostat combined with endocrine therapy may be an optional sequential strategy for patients with HR+/HER2-advanced breast cancer that has progressed on CDK4/6 inhibitor. In the revised manuscript, we have included the HDAC inhibitors in the new section 3.2.9.

2.2. Comment 2 (3.2.2. PI3K-AKT-mTOR inhibitors): It may be useful to mention CAPItello-291  phase III study evaluating the addition of AKT-inhibitor capivasertib to fulvestrant that showed a significant improvement in PFS in patients who had progressed to previous AI with or without CDK4/6 inhibitor (in February 2023, the FDA granted priority review designation to cabivasertib + fulvestrant) 

Once more, we do appreciate the excellent suggestion. In the revised manuscript, we have included this information in section 3.2.2.